# Multimedication Guidelines: Assessment of the Size of the Target Group for Medication Review and Description of the Frequency of Their Potential Drug Safety Problems with Routine Data

Veronika Lappe [1], Truc Sophia Dinh [2], Sebastian Harder [3], Maria-Sophie Brueckle [2], Joachim Fessler [4], Ursula Marschall [5], Christiane Muth [2,6], Ingrid Schubert [1,*] and on behalf of the EVITA Study Group [†]

1   PMV Forschungsgruppe, Faculty of Medicine and University Hospital Cologne, University of Cologne, 50931 Cologne, Germany; veronika.lappe@uk-koeln.de
2   Institute of General Practice, Goethe-University Frankfurt, 60590 Frankfurt am Main, Germany; dinh@allgemeinmedizin.uni-frankfurt.de (T.S.D.); brueckle@allgemeinmedizin.uni-frankfurt.de (M.-S.B.); christiane.muth@uni-bielefeld.de (C.M.)
3   Institute for Clinical Pharmacology, Medical Faculty, Goethe-University Frankfurt, 60590 Frankfurt am Main, Germany; harder@em.uni-frankfurt.de
4   General Practitioner, 65439 Floersheim, Germany; joachim.fessler@t-online.de
5   Head of Department Medicine/Health Care Research, BARMER, 42285 Wuppertal, Germany; ursula.marschall@barmer.de
6   Department of General Practice and Family Medicine, Medical Faculty OWL, University of Bielefeld, 33615 Bielefeld, Germany
*   Correspondence: ingrid.schubert@uk-koeln.de; Tel.: +49-221-478-85531
†   The Group Team is provided in the Appendix A.

**Abstract:** (1) Background: About 10 years ago, several guidelines for the better management of patients with polypharmacy were issued. A central issue is the definition of the target group. The primary aim of this study is therefore to assess the size of the target group, applying the criteria of the German guidelines. A further aim is to describe the frequency of occurrence of medication safety issues for patients of the target group. (2) Methods: The study is based on administrative data of one large statutory health insurer in Germany (*n* = 9,012,523). (3) Results: The criteria of multimorbidity (at least three chronic diseases) and utilization of five or more concurrent drugs over at least 91 days is fulfilled by 14.1% of the insured patients, or almost 1.3 million persons. About 5% of this multimorbid and poly-medicated population fulfilled at least three of out of five additional occasion-related criteria. Medication safety issues occur frequently: treatment prevalence with potentially inadequate medication, QT-drugs, benzodiazepine or Z-drugs and proton pump inhibitors was 30.4%, 28.9%, 11.1% and 52.4%, respectively. (4) Conclusions: The analysis shows the scope of patients eligible for a structured medication review and demonstrates the relevance for counselling based on the high percentage potentially at risk due to medication therapy safety problems.

**Keywords:** polypharmacy; medication review; general practice; guidelines; medication safety; drug utilization; administrative data

## 1. Introduction

About 10 years ago, several guidelines for the better management of patients with polypharmacy were issued: The German general practitioner guidelines "Multi-medication" from 2012 [1] with the updates (and upgrades to the highest evidence status S3) in 2021 [2], the Dutch guidelines "Multidisciplinaire Richtlijn Polyfarmacie bij ouderen" (2012) [3], the NICE guidelines "Medicines optimisation: the safe and effective use of medicines to enable the best possible outcomes" (2015) [4], "Multimorbidity: clinical assessment and

management" (2016) [5], and "Older people with social care needs and multiple long-term conditions" (2016) [6] and the guidelines of the American Geriatrics Society (AGS): "Guiding principles for the care of older adults with multimorbidity: an approach for clinicians" (2012) [7] are a few examples.

The motivation for the development of such guidelines was the intensified discussion about the management of the complex medication of the elderly and the difficulty of implementing numerous recommendations from various disease-specific guidelines. The pivotal paper of Boyd et al. (2005) [8] addressing this topic and studies on deprescribing [9,10] were widely acknowledged. The daily experience of a lack of overview of the patient's treatment and lack of clinical information for both patients and the treating physician was a further motivation for the German guideline development. This is partly due to the fact that in Germany patients are free to choose their family physician and other specialists in outpatient care. To date, an electronic patient file has not yet been introduced nationwide. Furthermore, there is no seamless care between the different sectors of care (outpatient, inpatient and nursing care), which can lead to a loss of information and breaks in continuity of care. In order to ensure the flow of information, at least with regard to medication, a medication plan has been introduced. Since October 2016, patients with statutory health insurance who take or use at least three systemically effective medicines at the same time over a longer period of time have been entitled to a nationwide standardized medication plan. This plan is predominantly used in paper-based form and is often not up to date [11]. The assessment and evaluation of the medication is therefore the core element of the guidelines. This leads to a central issue of the multimedication guidelines: the definition of the patient target group, i.e., the identification of those who might benefit from the management approach—but not all guidelines provide identification criteria. The NICE guidance on multimorbidity: clinical assessment and management [5], for example, recommends, besides a high number of prescribed drugs, some validated tools for identifying patients at risk for unplanned hospital admission or admission to care homes. Other selection criteria in guidelines were problems with adherence, treatment burden, drugs with high risk for adverse events, perceived limitations with the self-management of the therapy, social risk factors (e.g., lack of health literacy) and also treatment by multiple medical disciplines [2–5,12]. The German guidelines [2]—for the short version cf. Schubert et al. in *Pharmacoepidemiology* (2022, under review) [13]—has formulated two recommendations with regard to the target group: (1) Patients with polypharmacy (at least five concurrently used drugs) and multimorbidity (at least three chronic diseases) should undergo a medication review with inventory and assessment of medication at least once a year. (2) For patients with polypharmacy and multimorbidity with additional risks or events (e.g., falls, hospitalization), an occasion-related medication review (with medication inventory and assessment) should be carried out. Both recommendations have "B" as grade of recommendation, and V for level of evidence.

It is well known that there are multiple barriers to the implementation of guidelines. In addition to personal factors such as knowledge and attitude, external factors such as time constraints and lack of resources have to be considered [14,15]. So far, no information is available about how many patients would be eligible for a medication review according to the criteria of the German guidelines. Against this background, the primary aim of this study is therefore to calculate the size of the target group by applying these criteria of the German guidelines on health insurance data. Further aims are to describe the frequency of occurrence of some relevant medication safety issues for patients with polypharmacy and multimorbidity as the target group. This underlines the potential benefit of a medication review, as patients with polypharmacy are at increased risk for drug-related problems and adverse events [16].

This study is part of the EVITA project (Evidence-based Polypharmacy Program with Implementation in Health Care). Based on these results, the potential number of patients eligible for special care programs can be estimated.

## 2. Results

### 2.1. Prevalence of Polypharmacy and Multimorbidity

Overall, 14.1%, or almost 1.3 million, of people fulfilled the criteria for multimorbidity and polypharmacy over at least 91 days. Figure 1 presents data according to age and sex, and Table 1 shows the data for polypharmacy (5 and 10 concurrent drugs) over a longer time period (at least 182 and 272 days). As especially elderly patients are affected, the age group > 64 years is also presented (37.3%); men are more affected (40.5%) compared to women (35.3%).

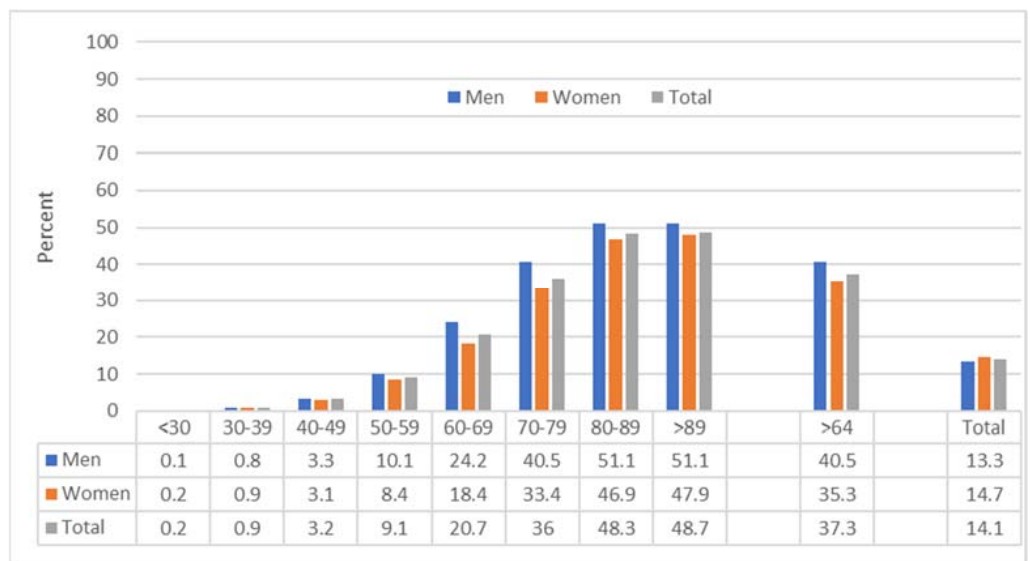

| | <30 | 30-39 | 40-49 | 50-59 | 60-69 | 70-79 | 80-89 | >89 | | >64 | | Total |
|---|---|---|---|---|---|---|---|---|---|---|---|---|
| Men | 0.1 | 0.8 | 3.3 | 10.1 | 24.2 | 40.5 | 51.1 | 51.1 | | 40.5 | | 13.3 |
| Women | 0.2 | 0.9 | 3.1 | 8.4 | 18.4 | 33.4 | 46.9 | 47.9 | | 35.3 | | 14.7 |
| Total | 0.2 | 0.9 | 3.2 | 9.1 | 20.7 | 36 | 48.3 | 48.7 | | 37.3 | | 14.1 |

**Figure 1.** Percentage of the insured persons with multimorbidity and polypharmacy, 2019. Continuously insured persons in 2019 or in the year before death, *n* = 9,012,523; multimorbidity and polypharmacy: at least three chronic diseases and at least five drugs prescribed simultaneously for at least 91 days during this period.

**Table 1.** Percentage of insured persons with multimorbidity and polypharmacy according to the number of concurrent drugs and duration, 2019.

| Percentage (%) of Insured Persons with Multimorbidity and Polypharmacy [1] | | | | | | |
|---|---|---|---|---|---|---|
| | Five or More Active Substances Over at Least | | | 10 or More Active Substances Over at Least… | | |
| Age group | 91 days | 182 days | 273 days | 91 days | 182 days | 273 days |
| <50 | 1.0 | 0.6 | 0.3 | 0.1 | 0.0 | 0.0 |
| 50–59 | 9.1 | 6.0 | 3.7 | 0.7 | 0.4 | 0.2 |
| 60–69 | 20.7 | 14.7 | 9.7 | 2.2 | 1.1 | 0.5 |
| 70–79 | 36.0 | 26.6 | 17.8 | 4.1 | 2.0 | 0.8 |
| 80–89 | 48.3 | 36.1 | 23.8 | 5.0 | 2.1 | 0.8 |
| >89 | 48.7 | 34.3 | 21.0 | 3.3 | 1.1 | 0.3 |
| <65 | 4.7 | 3.1 | 1.9 | 0.4 | 0.2 | 0.1 |
| >64 | 37.3 | 27.5 | 18.2 | 4.0 | 1.8 | 0.7 |
| 65–79 | 31.7 | 23.3 | 15.6 | 3.6 | 1.7 | 0.8 |
| >79 | 48.4 | 35.8 | 23.3 | 4.7 | 1.9 | 0.7 |
| Total | 14.1 | 10.2 | 6.6 | 1.4 | 0.7 | 0.3 |

[1] Continuously insured persons in 2019 or in the year before death, *n* = 9,012,523; multimorbidity and polypharmacy: at least three chronic diseases and at least 5 or 10 drugs used simultaneously for at least 91, 182 or 272 days during this period.

For those persons with multimorbidity and polypharmacy with at least five drugs used simultaneously for at least 91 days, the drug treatment prevalence (at least one prescription) was assessed. The five most common agents were pantoprazole (43.0%), hydrochlorothiazide (36.6%), metamizole (36.3%), amlodipine (32.8%) and ramipril (32.5%).

If we look at patients with at least one contact with a general practitioner in 2019, the proportion with multimorbidity and polypharmacy (five or more active agents over 91 days) amounts to 17.6% across all age groups; among those aged 65 years and older, this proportion is almost 40% (data not presented).

### 2.2. Target Group for Medication Review

In addition to the criteria of multimorbidity and polypharmacy, the following factors were analyzed, which, according to the guidelines of "multimedication", speak in favor of conducting a structured medication review: (i) patients with contact to three or more different physician groups, (ii) patients with two or more hospital stays in one year, (iii) patients with first time inpatient care, (iv) patients with psychotropic drug prescription and (v) patients with dementia:

Across all age groups, the analysis shows the following results for the patients with multimorbidity and polypharmacy (*n* = 1,272,351) (cf. Table 2):

- Almost 80% of the patients had contact with three or more different physician groups;
- In total, 17.4% had two or more hospitalizations;
- In total, 2.5% had inpatient nursing care for the first time;
- In total, 9.1% received at least one prescription for a psychotropic drug (men: 6.6%; women 10.8%);
- In total, 10% of the study population had a diagnosis of dementia or a corresponding prescription.

**Table 2.** Frequency of different criteria for identification of the target group for medication review, 2019.

| | | Percentage (%) of Insured Persons with Multimorbidity and Polypharmacy [1] and Utilization/Documentation of... | | | | |
|---|---|---|---|---|---|---|
| **Sex** | **Age Group** | **at Least Three Different Physician Groups** | **Two or More Hospitalizations** | **First time Inpatient Nursing Care** | **Psychotropics** | **Dementia** |
| Men | <65 | 69.7 | 13.8 | 0.4 | 11.8 | 1.2 |
| | 65–79 | 79.8 | 18.1 | 1.1 | 4.9 | 6.2 |
| | >79 | 81.2 | 23.6 | 4.5 | 4.6 | 19.5 |
| | total | 77.5 | 18.5 | 1.9 | 6.6 | 8.6 |
| Women | <65 | 82.5 | 12.8 | 0.4 | 18.0 | 1.0 |
| | 65–79 | 82.2 | 15.2 | 1.4 | 8.8 | 5.9 |
| | >79 | 72.4 | 20.4 | 6.0 | 8.9 | 22.3 |
| | total | 78.7 | 16.6 | 2.9 | 10.8 | 10.9 |
| Total | <65 | 76.6 | 13.2 | 0.4 | 15.1 | 1.1 |
| | 65–79 | 81.2 | 16.4 | 1.2 | 7.1 | 6.1 |
| | >79 | 75.4 | 21.5 | 5.5 | 7.4 | 21.4 |
| | total | 78.2 | 17.4 | 2.5 | 9.1 | 10.0 |

[1] Study population: insured persons with multimorbidity and polypharmacy, *n* = 1,272,351.

At least one of these five criteria was fulfilled by 85% of the study population with multimorbidity and polypharmacy, and three or more of these criteria by 5.4% (Figure 2).

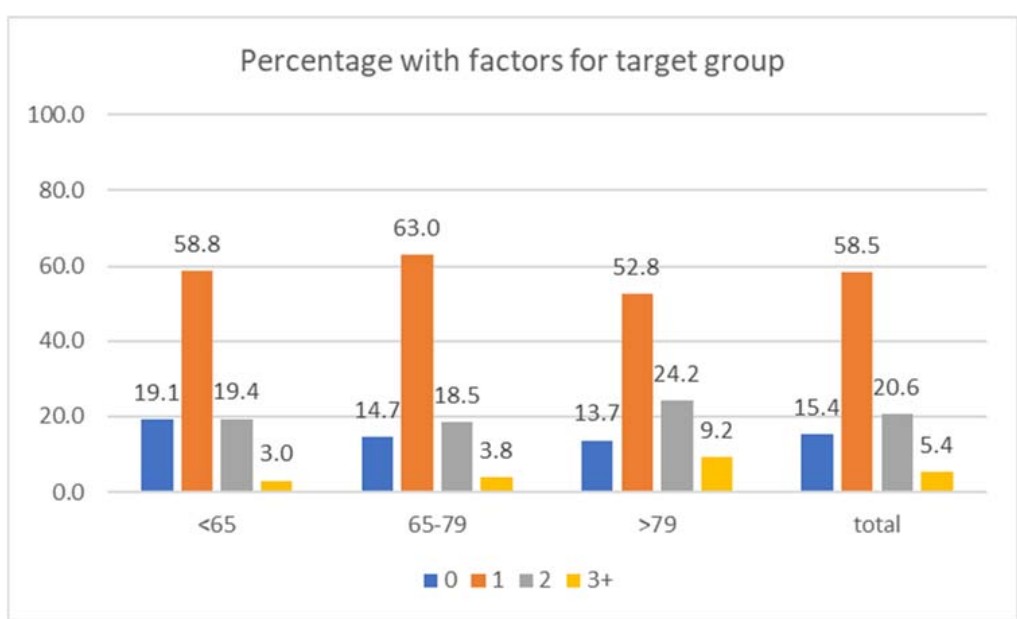

**Figure 2.** Percentage of the insured persons with multimorbidity and polypharmacy according to the number of potential factors as trigger for medication review. Study population: insured persons with multimorbidity and polypharmacy, *n* = 1,272,351.

### 2.3. Aspects of Medication Safety

Medication reviews aim to identify problems of the medication process and to optimize medication safety. Table 3 presents the frequency of selected problems that may interfere with the safe use of medicines that affect people with polypharmacy:

- About 30% of the population with multimorbidity and polypharmacy aged 65 years and older received at least one prescription with a drug classified as potentially inadequate for the elderly according to the PRISCUS list [17], the German PIM list. Better alternatives or special monitoring are recommended for these drugs.
- Overall, 29% (men: 25.7%, women: 31.1%) of the analyzed population with multimorbidity and polypharmacy received a drug with known drug-induced prolongation of the QT interval as a feared adverse and potentially life-threatening effect.
- Benzodiazepines, but also the so-called Z-drugs (zopiclone, zolpidem, and zaleplon), have a risk of drug dependence and should therefore only be prescribed for a short time. About 11% of the study population received at least one prescription of these agents (benzodiazepines: 7.7%; Z-drugs 4.5%).
- Across all age groups, treatment prevalence with PPI in the study population was around 52% in 2019; the mean daily dose was 442 DDD (around 8% received the therapy rather short-term, i.e., up to one quarter; around 18% of the insured persons received a prescription quantity that allowed for treatment with at least two DDD daily over the whole year (data not presented)).
- Almost 16% of the study population received a combination of ACE inhibitor/sartan/renin inhibitor plus diuretic plus non-steroidal anti-inflammatory drug (NSAID). This combination—called triple whammy—increases the risk of renal failure (cf. Table A2) [18].
- Drugs with anticholinergic side effects were prescribed to almost 26% of the population with multimorbidity and polypharmacy. Table 4 presents the data according to the degree of anticholinergic effects [19].

**Table 3.** Percentage of insured patients with multimorbidity and polypharmacy and at least one prescription of selected drug groups, 2019.

| | | | | | |
|---|---|---|---|---|---|
| **Percentage (%) of the Insured Patients with Multimorbidity and Polypharmacy [1] and at Least One Prescription of…** | | | | | |
| **Sex** | **Age Group** | **PIM \*** | **QT-Drug \*\*** | **Benzodiazepine/ Z-Drug \*\*\*** | **PPI \*\*\*\*** |
| Men | <65 | n.a. | 27.5 | 8.5 | 48.5 |
| | 65–79 | 23.7 | 24.0 | 7.3 | 47.0 |
| | >79 | 26.5 | 26.9 | 10.2 | 49.7 |
| | total | 24.8 | 25.7 | 8.4 | 48.2 |
| Women | <65 | n.a. | 37.5 | 12.6 | 55.1 |
| | 65–79 | 32.8 | 28.5 | 11.3 | 53.6 |
| | >79 | 35.5 | 30.4 | 14.9 | 57.5 |
| | total | 34.0 | 31.1 | 12.9 | 55.3 |
| Total | <65 | n.a. | 32.9 | 10.7 | 52.0 |
| | 65–79 | 28.9 | 26.6 | 9.6 | 50.8 |
| | >79 | 32.4 | 29.2 | 13.2 | 54.8 |
| | total | 30.4 | 28.9 | 11.1 | 52.4 |

[1] Study population insured with multimorbidity and polypharmacy *n* = 1,272,351, \* PIM—potentially inadequate medication according to Holt et al., 2011 [12]; \*\* cf. Table A1; \*\*\* ATC: N05BA, N05CF, \*\*\*\* Proton pump inhibitors—PPI: ATC A02BC.

**Table 4.** Frequency of anticholinergic drug prescriptions (according to anticholinergic risk scale of Rudolph et al. [19]) for the population with multimorbidity and polypharmacy, 2019.

| | | | | | | | |
|---|---|---|---|---|---|---|---|
| **Percentage (%) of the Insured Patients with Multimorbidity and Polypharmacy [1] with Anticholinergic Prescription \* According to Degree of Anticholinergic Effects** | | | | | | | |
| | **<65 Years** | | **65–79 Years** | | **>79 Years** | | **Total** |
| Degree of anticholinergic effects | men | women | men | women %  | men | women | |
| 1 | 17.7 | 24.8 | 13.7 | 19.0 | 17.5 | 24.7 | 19.7 |
| 2 | 4.7 | 5.4 | 3.3 | 3.5 | 3.0 | 3.0 | 3.7 |
| 3 | 6.1 | 10.9 | 3.7 | 6.8 | 3.9 | 6.5 | 6.3 |
| Total \*\* | 24.0 | 34.1 | 18.0 | 25.4 | 21.8 | 30.5 | 25.7 |

[1] Study population insured with multimorbidity and polypharmacy *n* = 1,272,351; 1 = weak effect; 3 = strong effect; \* The insured persons could have received anticholinergics of different degrees of anticholinergic effect; \*\* at least one drug with anticholinergic effect.

## 3. Discussion

### 3.1. Polypharmacy Prevalence

Polypharmacy occurs frequently, increases the risk of adverse events and is of public health concern due to the aging population. The study of Midão et al. (2018) presents polypharmacy prevalence for elderly (65 years and older) for 17 European countries and Israel—based on survey data [20]. The prevalence ranges from 26.3% to 39.9%; for Germany, an estimate of 30.3% (29.8–30.8) is reported. Our study is based on health insurance data and comprises all age groups. Applying the criteria of multimorbidity and at least five concurrent drugs for the target group resulted in 14.1% of the insured population being eligible for medication review. In the age group of 65 years and older, the percentage is around 37%, and in persons 80 years and older it is almost 50%. These figures are close to those reported by Midão et al., with 44.9% for persons 85 years and older [20]. As expected, at 1.4%, the percentage is much lower when using the criteria of at least 10 concurrent drugs for 91 days. Guidelines recommend further criteria such as frailty, vulnerability or treatment by multiple specialists for identifying persons who could benefit from a structured medication review [3,5,21]. In a recent French study [22] with a comparable database and inclusion of all age groups, the prevalence of chronic polypharmacy (five

drugs, 6 month) is 5.6%, compared to 10.2% in our study. For 10 or more drugs, the prevalences are comparable, with 0.5% in the French study and 0.7% in our study. Both studies show—as expected—an increase with age. The higher prevalence of our study might be due to different inclusion criteria of drug groups. Due to methodological differences in multimorbidity and polypharmacy studies [23,24], comparison is generally hampered.

### 3.2. Risk Situations and Occasions for Medication Review

The German guidelines for "multimedication" identify situations with an increased risk of uncoordinated prescribing, thus resulting in a greater lack of transparency of a patient's medication. Uncoordinated treatment is a health-system-related risk factor and is expected in patients with intensive healthcare utilization (ambulatory care, hospital care, long-term care). Therefore, they are included in the target population for medication review. Not surprisingly, almost 80% of the patients with multimorbidity and polypharmacy have seen three or more different specialists. This might lead to loss of information, double prescribing, problems with adherence and increased risk of interactions or other adverse events. Hospital stays are also an event where changes in the medication regime often occur, again with the risk of inadequate communication with patients and office-based physicians. Therefore, guidelines recommend taking this as an occasion for medication review. A further occasion for medication review is the change in the status of a person from 'no nursing care' to 'first time institutionalized nursing care', as drug-related problems could be involved. This affected 2.5% of the patients with multimorbidity and polypharmacy. In general, care recipients have a high risk of polypharmacy: 29% of our study population received nursing care [25]. Especially in long-term care facilities, polypharmacy is high and of concern, as mostly older and frail persons are affected [26].

Disease and medication-related risk factors also serve as pick-up criteria for medication review for psychotropic drugs in general, but also a number of psychotropics are associated with increased risk for fall injuries, hospitalization and death. Patients with dementia are especially vulnerable [27]. Due to a high risk of interactions and adverse events, but also of adherence problems, multimedication guidelines recommended reviewing the medication of those with psychotropic drugs and dementia on a regular basis (7.4% and 10%, respectively, in our study population).

About 5% of the multimorbidity and polypharmacy population fulfilled three or more of these occasion-related criteria. These estimates show that the definition of the target group (and the intervals at which medication reviews should take place) have a direct impact on the time and financial resources of the doctor's practice. For the Dutch guideline "Polypharmacy" (2012) [3], a budget impact analysis was published in 2020 [28]. In general practice, 20% with a range of 12–30% met the criteria of polypharmacy (55 years and older and more than five drugs). The number of patients at high risk of pharmacotherapy, defined as patients aged 75 years and older with chronic use of $\geq 10$ medications and/or with demonstrated susceptibility, is estimated by the authors at 62 in a standard practice and 97 in an average-sized general practice. No comparable data are available for Germany. Based on the data of our study, in a general practice with 1200 patients per quarter, about 211 patients are expected to fulfill the criteria of multimorbidity and polypharmacy with five or more concurrent drugs over 91 days. Of those, 11 patients fulfill three or more occasion-related criteria. The number of patients is of course influenced by the age structure of the general practitioner's clientele.

### 3.3. Medication Safety Issues in Patients with Polypharmacy

The further analysis of selected aspects of medication safety demonstrates the relevance for medication reviews and shows the potential for changes or deprescribing for the identified target group:

- The prescription of PIM is analyzed in many countries [29,30], though changes in everyday practice seem to be difficult [31,32]. Overall, 30% of the study population had at least one prescription classified as PIM, with a higher percentage of women

compared to men, among other reasons due to the higher prescribing of psychotropics in women [33].

- The risk of QT interval prolongation increases in the presence of multimedication [34]. The frequency of prescribing drugs that prolong the QT interval might be underestimated by practitioners due to lack of information. In the population under study, almost 29% received least one non-cardiac drug with this potential adverse drug effect. The prevalence estimate is influenced by the population studied and the drugs included. Comparison with other studies is hampered by differences in the study population and numbers of drugs included. Schaechtele et al. report for a cohort of 130,434 geriatric patients (mean age 81 years, 67% women) a treatment prevalence of 58.7% for drugs associated with any QT risk (ALL-QT) and 17.2% for drugs classified as a high-QT-risk drug [35]. Our analysis was restricted to non-cardiac drugs. The German guidelines for "multimedication" urges having an ECG performed in patients with QT-time-prolonging drugs.

- Benzodiazepine or Z-drugs with their risk of increasing falls and drug dependence are often prescribed for too-long periods. Prevalence estimates are not only influenced by the characteristics of the population studied, but also by reimbursement schemes or prescribing recommendations. A recently published study analyzed the use of benzodiazepines and other hypnotics in multimorbid older people ($\geq$2 chronic diseases) in the province of Quebec (Canada) with routine data [36]. Despite a decline since 2000, almost 31% still received at least one prescription in 2016. A study from Spain [37] reports a prevalence of 28.4% for patients with polypharmacy—a joint prescription of five or more drugs—in 2015. The prevalence in our study is much lower (7.7%), which could partly be due to private prescribing not included in routine data or a more restrained prescription.

- Potential for deprescribing is not only seen for benzodiazepine or Z-drugs, but also for proton pump inhibitors. Internationally, this problem of overprescribing has been also addressed [38–41]. The above-cited study from Spain identified PPI as the most prescribed drug with 63.3% (2015) [37]. PPIs were also prescribed most frequently in our study: 52% received at least one prescription. Medication review helps to detect the correctness of indication.

- Drug combinations such as renin inhibitors together with diuretics and NSAIDs increase the risk of renal failure [18] and should be avoided. In the presented analysis, 16% of the study population received this combination, which demonstrates, among other measures, the necessity for medication reviews. The percentage of patients at risk is underestimated in our analysis, as NSAIDs in self-medication could not be included.

- Anticholinergic drugs are an important issue for the medication review, as they can lead to a variety of health problems and adverse drug effects, especially in the elderly [42]. During medication review, physicians should discuss alternatives with the patients. According to the risk scale of Rudolph et al. (2008) [19], 25% of the study population received at least one drug, and about 6% received a drug with strong anticholinergic effects. Comparable to the analysis of QT drugs, prevalence estimates are influenced by the characteristics of the population under study and the drugs or risk scale included. Krueger et al. (2021) reported a prevalence of anticholinergic drug use according to the list of Carnahan et al. (2006) [43] of 38.4%, and 53.7% applying the German list of Kiesel et al. (2018) [44] for a patients with at least three diagnosed chronic diseases and aged between 65 and 85 years [45]. The authors point out the need to be aware of the cumulative effects of the use of several lower potential anticholinergic drugs, which in turn leads to high anticholinergic burden.

### 3.4. Strength and Limitation

To the best of our knowledge, this is the first study that assessed the size of the target group for medication review defined by the German guidelines "multimedication". The

study is based on a large SHI population. In addition to the size of the administrative database, further strengths are seen in the fact that no dropout or recall bias occurs. Another strength is that we used a stricter definition of polypharmacy compared to many other studies, with the requirement of overlapping use of five or more different active substances over at least 91 days.

Insurance funds differ in their clientele. The BARMER has a somewhat higher percentage of elderly and female insurees, but according to surveys, comparable health status of the persons insured [46]. Therefore, we cannot rule out that we overestimate the number of eligible persons due to the age and gender differences. Further limitations are seen in the fact that multimorbidity was defined by the documentation of three or more ICD-10 diagnosis groups according to the guidelines. There is no international standard for the definition of multimorbidity. It has to be taken into consideration that our results are based on the multimorbidity definition that is mostly used (three or more chronic diseases), and results based on other definitions would differ. Misclassification of diagnoses cannot be ruled out in routine data, but for our study, the number of diagnostic groups are relevant, not the single diagnosis. Therefore, we are confident that the results are robust.

For the assessment of polypharmacy, only prescribed drugs could be included as we have no information about self-medication. In addition, we have no information on whether at all (non-adherence) or in what way (irregular) the prescribed drugs have been taken, but we know that the doctors have prescribed these drugs and the patient has collected the medication in a pharmacy and is therefore a patient with intended multimedication, which should be regularly reviewed. Medication safety was limited to selected issues. Further studies should also focus on the underuse of drugs.

## 4. Materials and Methods

Database: The study is based on the administrative data of one of the largest statutory health insurance (SHI) companies in Germany. The BARMER covers a population of about 12% (9 million persons) of the insured population. About 88% (73 million) of the German population (83 million) are insured with one of the 103 statutory health insurance funds. Since 1993, there is freedom to choose the insurance fund. The German healthcare system is based on the Bismarck model of a social health insurance system with its principal of solidarity (all insured persons pay a certain percentage of their income regardless of their health status) and self-governance (for an overview cf. [47]) The BARMER is one of the largest health insurers and the data have been used for many years for health service research.

Data were accessed via the science data warehouse (W-DWH) of the BARMER, providing anonymized data of all sectors of care—e.g., ambulatory data with physician-coded diagnosis according to International Statistical Classification of Diseases and Related Health Problems 10th Revision for Germany (ICD 10-GM), drug prescription with information on the kind of drug according to the anatomical–therapeutic–chemical (ATC) code and defined daily doses (DDD), hospital stay with date and ICD-10-diagnosis, statutory nursing care) as well as year of birth and sex of the insured person.

Basic population: All persons insured continuously in 2019, and those who died in 2019 and were insured continuously in the year prior to their death, were included (*n* = 9,012,523).

Observation period: Morbidity, medication and utilization of outpatient and inpatient treatment as well as care services of the insured persons were examined for the year 2019, or in case of death, in 2019, for the year before date of death.

Definition of multimorbidity and polypharmacy: A person with multimorbidity was defined by the documentation of three or more chronic diseases in 2019. Diseases were the 241 disease groups in the chapters of the ICD-10 code, German modification [48]. In these disease groups, three-digit ICD codes are summarized, e.g., ICD codes E10 to E14 are diabetes. Chapter XVIII of the ICD code "Symptoms, signs and abnormal clinical and laboratory findings, not elsewhere classified" and chapter XXI "Factors influencing health

status and contact with health services" were omitted. A disease was regarded to be chronic if the disease group was documented in three of four quarters in 2019; thereby, codes within the group could change. The outpatient diagnoses marked as "confirmed" as well as the inpatient discharge diagnoses of the insured persons in 2019, or, for those who died in 2019, diagnoses in the four quarters before the quarter of death, were taken into account.

Polypharmacy: The analysis was based on drug prescribing in the outpatient sector. Polypharmacy was operationalized as the concomitant use of five or more (high level: 10 or more) agents (ATC 7-digit) during a period of at least 91 days. This time window was chosen to include patients with chronic use of polypharmacy over a substantial period and to exclude patients with short-term use. Combinations were broken down according to their individual active substances. Of the ATC group Varia (ATC code "V"), only the subgroup "V03", "All other therapeutic agents" without tissue adhesives, agents for embolization, medical gases and ethanol, was included. Dermatologicals (ATC code "D") were excluded from the evaluation on polypharmacy because their systemic effects cannot be estimated in claims that do not contain data on dosage. Overlapping ranges of the different drugs to assess concurrent use were calculated on the basis of the defined daily dose (DDD) with the assumption that the intake of the medicine started with the pharmacy dispensing date.

Study population for drug safety analysis: Insured persons fulfilling the criteria of multimorbidity ($\geq$3 chronic diseases) and polypharmacy with 5 or more concurrent active substances (ATC 7-digit) over at least 91 days (*n*: 1,272,351).

Study measures: The multimedication guideline identifies occasions when doctors should consider a medication review. These occasions (here: study measures) were operationalized with routine data as follows in order to identify the number of people affected.

- Contact with different physician groups: Physician groups are identified by the lifetime doctor's number (8th and 9th position of the 9-digit code). Diagnosing physicians such as microbiologists and radiologists, as well as pathologists and unknown specialty were omitted. Different subgroups of a specialty were combined into a group.
- Number of hospital stays: Hospital stays with at least one overnight stay in the observation year 2019 were included. Overlapping or immediately following hospital stays were counted as one hospital stay.
- First time of inpatient nursing care: First-time inpatient care was assumed if patients with multimorbidity and polypharmacy had only outpatient care or no care in the 92-day quarter prior to their first day of inpatient care in the observation year.
- Prescribing of psychotropic drugs: prescriptions with ATC Code N05 and N06, excluding N05BP, N05CP, N05H, N06AH, N06AP (= herbal or homoeopathic drugs) and N06D (= antidementia drugs).
- Patients with dementia: Documentation of the ICD-10 diagnoses G30 (Alzheimer's disease) or F00 to F03 (dementia) or at least one prescription of an antidementia drug (ATC code N06D)
- Frequency of drug utilization: Potentially inadequate medication according to PRISCUS list [12], QT-drugs (Table A1), benzodiazepines and Z-drugs (ATC codes: N05BA, N05CF), proton pump inhibitors (ATC codes A02BC), "triple whammy" combination: prescribing of agents acting on the renin–angiotensin system (ATC-code: C09A or C09C, or C09XA and diuretics (ATC codes: C03 or C02L or C07B/-C/-D or C08G) and antiphlogistics (ATC codes: M01A und NSAID in other groups).

Statistics: The study applies descriptive methods to assess the number of patients eligible for a medication review fulfilling the criteria according to the recommendation in the German guidelines "multimedication". The denominator is given by all persons continuously insured and one year continuously insured before death, respectively (*n* = 9,012,523), and the nominator by those fulfilling the definition of multimorbidity and polypharmacy (see above). We further calculated the percentage of insured persons with certain study measures (see above) related to the number of those fulfilling the criteria of measures (nominator) related to the number of the study population with multimorbidity and polypharmacy (*n* = 1,272,351) as the denominator. We followed guidelines for the work

with administrative data [49]. To analyze the data, the Microsoft SQL Server Management Studio (SSMS) v18.7.1 was used.

## 5. Conclusions

The analysis carried out here shows the relevance of a structured medication review based on the scope of patients potentially at risk due to multimedication and medication therapy safety problems. A prerequisite for the improvement of medication therapy safety is transparency about all the drugs used by the patient, as well as knowledge of their diseases and their utilization of healthcare services for all healthcare professionals involved. A medication plan is the first step, followed by regular medication review for patients at risk of medication-related problems and high disease burden.

**Author Contributions:** Conceptualization: V.L. and I.S.; methodology: V.L. and I.S.; formal analysis: V.L.; writing—original draft preparation, I.S. and V.L.; writing—review and editing: T.S.D. and M.-S.B.; supervision: J.F., S.H., U.M. and C.M.; funding acquisition: C.M. and I.S. All authors have read and agreed to the published version of the manuscript.

**Funding:** This research as part of the EVITA study (Evidence-based Polypharmacy Program with Implementation in Health Care) was funded by the German Innovation Funds of the German Joint Federal Committee, grant number 01VSF16034.

**Institutional Review Board Statement:** Not applicable.

**Informed Consent Statement:** Not applicable.

**Data Availability Statement:** Data are presented in the article and Appendix A.

**Acknowledgments:** The authors would like to thank the members of the Guideline Group of General Practitioners in Hesse, Dres. Braun, Grenz, Graafen, Huettner, Meissl, Reincke, Seffrin, Vetter, and Beyer (†) and the BARMER for the possibility to analyze the data of the insurance.

**Conflicts of Interest:** V.L., T.S.D., M.-S.B., I.S. and C.M. received funding by the Innovation Fund for this project. J.F. works in his own office as a general practitioner, U.M. is employed by the BARMER, S.H by the Institute for Clinical Pharmacology, Medical Faculty of the Goethe University, Frankfurt/M. All authors were involved in the update and upgrade of the polypharmacy guideline. The authors declare no conflict of interest. The funder had no role in the design of the study; in the collection, analyses, or interpretation of data; in the writing of the manuscript, or in the decision to publish the results.

## Appendix A

**Table A1.** QT drugs: selected drugs with non-cardiac indication [50].

| Indication | Active Substances (Examples) |
| --- | --- |
| Antibiotics | azithromycin, clarithromycin, erythromycin, ciprofloxacin, levofloxacin, moxifloxacin, ofloxacin, trimethoprim-sulfamethoxazole |
| Antidepressants | amitriptyline, citalopram, escitalopram, doxepin, fluoxetine, imipramine |
| Antiemetics | granisetron, ondansetron |
| Antihistamines | terfenadine |
| Antimycotics | fluconazole, ketoconazole |
| Antipsychotics | chlorpromazine, clozapine, droperidol, fluphenazine, haloperidol, olanzapine, pimozide, paliperidone, quetiapine, risperidone |
| Antiasthmatic agents | salbutamol, salmeterol, terbutaline |
| Antiprotozoals | quinine, chloroquine, mefloquine, pentamidine |
| other | amantadine, amiodaron, alfuzosin, octreotide, tacrolimus, tamoxifen, vardenafil |

**Table A2.** Number and percentage of persons with multimorbidity and polypharmacy and prescription of "triple whammy" combination.

| Drug Group | Number and Percentage (%) of Study Population [1] with Prescription | |
| --- | --- | --- |
| | *n* | % |
| ACE inhibitor/sartan/renin inhibitor | 1,028,867 | 80.9 |
| plus diuretic * | 730,253 | 57.4 |
| plus diuretic and NSAID * | 199,992 | 15.7 |

[1] Insured persons with at least three chronic diseases and at least five drugs used simultaneously for at least 91 days in 2019 Total *n* = 1,272,351; * concomitant use possible for at least 5 days.

**EVITA Study Group:** Maria-Sophie Brückle [1], Truc Sophia Dinh [1], Ferdinand Michael Gerlach [1], Ana Isabel González-González [1], Wolfgang Greiner [2], Tobias Haber [3], Veronika Lappe [4], Ursula Marschall [5], Christiane Muth [1,6], Maximilian Pilz [7], Achim Schmidtko [8], Ingrid Schubert [4], Manfred Schubert-Zsilavecz [9], Svenja Seide [7], Marjan van den Akker [1], Julian Witte [2].

[1] Institute of General Practice, Goethe-University Frankfurt, Theodor-Stern-Kai 7, 60590 Frankfurt am Main, Germany; [2] Department of Health Economics and Health Care Management, Faculty of Health Science, Bielefeld University, Universitaetsstr. 25, 33615 Bielefeld, Germany; [3] INSIGHT Health GmbH & Co. KG, Auf der Lind 10a, 65529 Waldems-Esch, Germany; [4] PMV Research Group, Faculty of Medicine and University Hospital Cologne, University of Cologne, Cologne, Germany; [5] Head of Department Medicine/Health Care Research, Barmer, Lichtscheider Str. 89, 42285 Wuppertal, Germany; [6] Department of General Practice and Family Medicine, Medical Faculty OWL, University of Bielefeld, Universitaetsstrasse 25, 33615 Bielefeld, Germany; [7] Institute of Medical Biometry and Informatics, Heidelberg University Hospital, Im Neuenheimer Feld 130.3, 69120 Heidelberg, Germany; [8] Institute of Pharmacology and Clinical Pharmacy, Goethe University, Max-von-Laue-Str. 9, 60438, Frankfurt am Main, Germany; [9] Institute of Pharmaceutical Chemistry/ZAFES, Goethe University, Max-von-Laue-Str. 9, 60438, Frankfurt am Main, Germany.

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
