# Peer review of "Multimedication Guidelines: Assessment of the Size of the Target Group for Medication Review and Description of the Frequency of Their Potential Drug Safety Problems with Routine Data"

_2813-0618, doi:10.3390/pharma1010002_

Round 1

Reviewer 1 Report

  1. Abstract: Some parts are not very informative. The almost 1.3 million persons is only a useful piece of information when the denominator is known.
  2. Why are there those numbers like “(1) Background”, “(2) Methods” and “(3) Results”?
  3. Introduction: The introduction gives a good outline on the relevance of guidelines on polypharmacy. However, the transition to the aim of this study is somewhat weak as a research gap and a discussion on the current literature on the prevalence of polypharmacy is lacking.
  4. Methods: In general, the data used are well described. What is somewhat unclear is how the definition of multimorbidity (three or more chronic diseases in at least three quarters) was applied. Was any ICD code chronic as long as it was coded in at least three quarters or was there a list? On which level of the ICD code was this done (e.g. was diabetes counted once as long as E10-E14 was coded or were E11 and E14 counted twice or was even E14.8 and E14.9 counted twice)? Please provide more information on that.
  5. The authors write “Combinations were broken down according to their individual active substances.” Was this done for any combination (including asthma sprays or eye drops) or only for oral preparations?
  6. Number of hospital stays: How was this counted with respect to overlaps (e.g. was a stay ending on one day and another one starting on the same day counted as one or two hospitalizations)?
  7. The authors only assessed nursing home stays. Why didn’t they assess persons being care dependent as a whole?
  8. Overall, the methods section reads more like a string of bullet points than like a flowing text. Furthermore, a clear description of what is analyzed (a section on statistical analysis) is lacking. This is important as the plan what was done should be made transparent in the methods.
  9. Results: The statement “The most common agents used in polypharmacy were pantoprazole, hydrochlorothiazide, metamizole, amlodipine and ramipril.” should include more information (e.g. percentages and was that the number of prescriptions or DDD).
  10. Section 3.3 (Aspects of Medication Safety) starts with a justification and a reference to the PRISCUS list. This should not be part of the results.
  11. Table 4 is in a different format as it is the only table also including numbers and percentages. All tables should have the same style.
  12. Discussion: Some parts of the discussion are also written in bullet points. Several points are discussed within a few sentences and do not follow a clear story. The text must be better structured, the paragraphs must build on each other and consequences of the findings should be deduced.
  13. Another limitation might be medication prescribed or taken as needed.
  14. Conclusion: Interestingly but somewhat unexpected, medication plans are mentioned for the first time in the last sentence of the manuscript. The role and the potential of medication plans should already be discussed above.

e paper.

Author Response

Dear Reviewer,

the authors would like to thank you for their time and helpful comments that we addressed as follows. The changes in the manuscript are presented in italic. We changed the structure of the manuscript according to specification of the editor

Review 1

Comments and Suggestions for Authors

  1. Abstract: Some parts are not very informative. The almost 1.3 million persons is only a useful piece of information when the denominator is known.
    Answer: The percentage was provided (14.1%), but we added the number of the study population (n= 9,012,523) in the methods section. Due to the word count, we did not include further results.

  1. Why are there those numbers like “(1) Background”, “(2) Methods” and “(3) Results”?
    Answer: The numbers were already given in the template of the journal.

  1. Introduction: The introduction gives a good outline on the relevance of guidelines on polypharmacy. However, the transition to the aim of this study is somewhat weak as a research gap and a discussion on the current literature on the prevalence of polypharmacy is lacking.

Answer: Thank you for this feedback and the suggestions. We integrated a new paragraph explaining the background of our research aim (see below). The discussion on the current literature on polypharmacy prevalence was now added in our discussion section (see changes in the manuscript)

It is well known that there are multiple barriers to the implementation of guidelines and their recommendations. Beside personal factors like knowledge and attitude, external factors such as time constraints and lack of resource have to be considered [14, 15]. So far, no information is available, how many patients would be eligible for a medication review according to the criteria of the German guideline. Against this background, the primary aim of this study is therefore to calculate the size of the target group by applying these criteria of the German guidelines on health insurance data. Further aims are to describe the frequency of occurrence of some relevant medication safety issues for patients with polypharmacy and multimorbidity as target group. This underlines the potential benefit of the medication review, as patients with polypharmacy are at increased risk for drug related problems and adverse events [16].

Besides, we integrated some further information related to the German health care system (see Reviewer 2)

  1. Methods: In general, the data used are well described. What is somewhat unclear is how the definition of multimorbidity (three or more chronic diseases in at least three quarters) was applied. Was any ICD code chronic as long as it was coded in at least three quarters or was there a list? On which level of the ICD code was this done (e.g. was diabetes counted once as long as E10-E14 was coded or were E11 and E14 counted twice or was even E14.8 and E14.9 counted twice)? Please provide more information on that.

Answer: We added some details to the definition of multimorbidity. Any code was used except chapter XVIII and XXI of the ICD code. The groups of the chapters which describe diseases or disease groups were used thereby; codes within the group could change.

The paragraph is now written as follows:

Diseases were the 241 disease groups in the chapters of the ICD-10 code, German modification
. In these disease groups three-digit ICD codes are summarized, e.g. ICD codes E10 to E14 is diabetes. Chapter XVIII of the ICD code “Symptoms, signs and abnormal clinical and laboratory findings, not elsewhere classified” and chapter XXI “Factors influencing health status and contact with health services” were omitted. A disease was regarded to be chronic if the disease group was documented in three of four quarters in 2019 thereby codes within the group could change.

  1. The authors write “Combinations were broken down according to their individual active substances.” Was this done for any combination (including asthma sprays or eye drops) or only for oral preparations?
    Answer: All combinations were broken down.

  2. Number of hospital stays: How was this counted with respect to overlaps (e.g. was a stay ending on one day and another one starting on the same day counted as one or two hospitalizations)?

Answer: We added in the text: Overlapping or immediately following hospital stays were counted as one hospital stay.

  1. The authors only assessed nursing home stays. Why didn’t they assess persons being care dependent as a whole?

Answer: The aim of the study was to assess the size of the population which form the target group for medication review as recommended in the guidelines. One criterion is the beginning of a nursing home stay. Medication should be reviewed to rule out the possibility that a drug-related problem may have led to a worsening of the condition. For the readers we added the percentage of those with nursing care as a background information in the discussion section. We added the following sentences in the second paragraph of the section “Discussion”:

A further occasion for medication review is the change in the status of a person from ‘no nursing care’ to ‘first time institutionalized nursing care’ as drug related problems could be involved. This affected 2.5% of the patients with multimorbidity and polypharmacy. In general, care recipients have a high risk for polypharmacy. 29% of our study population received nursing care [28]. Especially in long-term care facilities polypharmacy is high and of concern, as mostly older and frail persons are affected [29].

  1. Overall, the methods section reads more like a string of bullet points than like a flowing text. Furthermore, a clear description of what is analyzed (a section on statistical analysis) is lacking. This is important as the plan what was done should be made transparent in the methods.

    Answer: Thank you for the hints. We added a sentence to explain the descriptive analysis of our measures. As many operationalizations are necessary for our study measures we prefer the bullet point presentation for better orientation of the readers. We now give a short introduction.

Study measures: The multimedication guideline identifies occasions when doctors should consider a medication review. These occasions (here: study measures) were operationalized with routine data as follows in order to identify the number of people affected.

Additionally, we included a short paragraph concerning the statistic.

Statistics
The study applies descriptive methods to assess the number of patients eligible for medication review fulfilling the criteria according to the recommendation in the German guidelines “multimedication”. The denominator is given by all persons insured (N= 9,012,523), the nominator by those fulfilling the definition of multimorbidity and polypharmacy (see above). We further analyze frequencies of different study measures (see above) as the number of those fulfilling the criteria of the measures (nominator) related to the number of the study population with multimorbidity and polypharmacy (N = 1,272,351) as denominator.

  1. Results: The statement “The most common agents used in polypharmacy were pantoprazole, hydrochlorothiazide, metamizole, amlodipine and ramipril.” should include more information (e.g. percentages and was that the number of prescriptions or DDD).

Answer: We added the percentages and the operationalization of the measure (persons with at least one prescription of the respective drug). We changed the position of this paragraph and have inserted it after table 1.

For those persons with multimorbidity and polypharmacy with at least 5 drugs used simultaneously for at least 91 days, the drug treatment prevalence (at least one prescription) was assessed. The five most common agents were pantoprazole (43.0%), hydrochlorothiazide (36.6%), metamizole (36.3%), amlodipine (32.8%) and ramipril (32.5%).

  1. Section 3.3 (Aspects of Medication Safety) starts with a justification and a reference to the PRISCUS list. This should not be part of the results.

    Answer: There seems to be a misunderstanding here. The potentially inadequate medication according to the drugs of the PRISCUS list (the German PIM list) is a study measure as described in the method section. To avoid misunderstanding we changed the wording of the sentence. The sentence now reads as follows:

    About 30% of the population with multimorbidity and polypharmacy aged 65 years and older received at least one prescription with a drug classified as potentially inadequate for elderly according to the PRISCUS list [15], the German PIM list. Better alternatives or special monitoring are recommended for those drugs.

  1. Table 4 is in a different format as it is the only table also including numbers and percentages. All tables should have the same style.

    Answer: Thank you for this hint. We adapted the format according to table 3.

  1. Discussion: Some parts of the discussion are also written in bullet points. Several points are discussed within a few sentences and do not follow a clear story. The text must be better structured, the paragraphs must build on each other and consequences of the findings should be deduced.

Answer: Thank you for this feedback. We now integrated headlines for better orientation of the intended structure. We kept the bullet points for the medication safety issues according to the result section. For each issue, we shortly discuss the result. All changes are marked in the manuscript.

  1. Another limitation might be medication prescribed or taken as needed.

Answer: We are not quite sure, whether we understood this comment. We mentioned already in the limitations that we have no information if at all or in what manner prescribed drugs have been taken. We only know that they were collected from the pharmacy and we assume the application of the medication. From the perspective of the treating physicians, it is a patient with polypharmacy and the medication should be checked on a regular basis, beside others to uncover non-adherence or other drug related problems. We changed the sentence in the strength and limitations section as follows.

In addition, we have no information on whether at all (non-adherence) or in what way (irregular) the prescribed drugs have been taken, but we know that the doctors have prescribed these drugs and the patient has collected the medication in a pharmacy and is therefore a patient with intended multimedication, which should be regularly reviewed.

  1. Conclusion: Interestingly but somewhat unexpected, medication plans are mentioned for the first time in the last sentence of the manuscript. The role and the potential of medication plans should already be discussed above.

Answer: Thank you for this comment. We now mention the medication plan in the first section with some further information on the German health care system. We added the following paragraph

The daily experience of a lack of overview of the patient’s treatment and lack of clinical information for both, patients and treating physician, was a further motivation for the guideline development. This is partly due to the fact, that in Germany patients are free to choose their family physician and other specialists in outpatient care. To date, the electronic patient file has not yet been introduced nationwide. Furthermore, there is no seamless care between the different sectors of care (outpatient, inpatient care and nursing care), which can lead to a loss of information and breaks in continuity of care. In order to ensure the flow of information, at least with regard to medication, a medication plan has been introduced. Since October 2016 patients with statutory health insurance who take or use at least three systemically effective medicines at the same time over a longer period of time have been entitled to a nationwide standardized medication plan. This plan is predominantly used in paper-based form and often not up to date [11]. The assessment and evaluation of the medication is therefore the core element of the guidelines.  

Reviewer 2 Report

The aim of this study is  to estimate the size of the target group and to describe the frequency of occurrence of medication safety issues for patients of the target group in Germany. This study  revealed  the potential number of insured persons who belong to the target group of a structured medication review in Germany. There are several major issued to be published in the followings.

(1) The authors used the terminology of "estimate", however, their analysis is mainly descriptive analysis in the current version of the  draft. If the authors would like to discuss about the "estimate" of the multi-morbidity and polypharmacy state in Germany, the authors should consider the appropriate statistical methods to estimate its prevalence with 95% confidence interval. 

(2) The authors should define the study design in the Material and Methods. Is it cross-sectional study or descriptive study?  In addition, this study population is all insurers in 2019, however, this study’s purpose is multi-morbidity and polypharmacy, so the target population could be elderly population or possibly including middle aged population. It is not clear why the author did not have any criteria to define the population, such as age.

(3) The authors should show the information about database appropriateness, such as generalizability of this database. This journal is an international journal, and the author should provide a short explanation about German insurance system to the readers.

(4) The authors defined for the criteria of multi-morbidity (≥3 chronic diseases) and polypharmacy with 5 or more concurrent active substances (ATC 7-digit) over at least 91 days. Please explain the reason for the cutoff points. 

(5)This study  aims are to describe the frequency of occurrence of some medication safety issues. The authors should analyze and discuss about the risk of adverse effects on the polypharmacy. In addition, in Table 3, the authors should described about something related to the multi-morbidity status and the polypharmacy in this study population. 

(6) The recommendation is to  analyze safety issues for the polypharmacy in the multiple model analysis, since only prevalence analysis results in influenced by confounding effects, and it would be not appropriate to conclude this study purpose.

(7) The study limitation is not enough discussed. The authors should discuss about the non-adherence issues. Further whether claim data has no misclassification error should be discussed. 

(8) The gender difference should be observed in terms of the combination of polypharmacy and multi-morbidity issues. 

Author Response

Review 2

Dear Reviewer,

the authors would like to thank you for their time and helpful comments that we addressed as follows. The changes in the manuscript are presented in italic. We changed the structure of the manuscript according to specification of the editor

Comments and Suggestions for Authors

The aim of this study is  to estimate the size of the target group and to describe the frequency of occurrence of medication safety issues for patients of the target group in Germany. This study  revealed  the potential number of insured persons who belong to the target group of a structured medication review in Germany. There are several major issued to be published in the followings.

(1) The authors used the terminology of "estimate", however, their analysis is mainly descriptive analysis in the current version of the  draft. If the authors would like to discuss about the "estimate" of the multi-morbidity and polypharmacy state in Germany, the authors should consider the appropriate statistical methods to estimate its prevalence with 95% confidence interval. 

Answer: Thank you for this comment. As the reviewer correctly notes, the study is descriptive. We assessed the number of persons fulfilling certain criteria in the population of one health insurance as full census, therefore we have no 95%-CI. We now avoid the word “estimate”. The title reads as follows:
Multimedication Guidelines: Assessment of the Size of the Target Group for Medication Review and Description of the Frequency of Their Potential Drug Safety Problems with Routine Data

(2) The authors should define the study design in the Material and Methods. Is it cross-sectional study or descriptive study? 
Answer: We added a section on the statistical method.

Statistics
The study applies descriptive methods to assess the number of patients eligible for medication review fulfilling the criteria according to the recommendation in the German guidelines “multimedication”. The denominator is given by all persons insured (N= 9,012,523), the nominator by those fulfilling the definition of multimorbidity and polypharmacy (see above). We further analyze frequencies of different study measures (see above) as the number of those fulfilling the criteria of the measures (nominator) related to the number of the study population with multimorbidity and polypharmacy (N = 1,272,351) as denominator.

In addition, this study population is all insurers in 2019, however, this study’s purpose is multi-morbidity and polypharmacy, so the target population could be elderly population or possibly including middle aged population. It is not clear why the author did not have any criteria to define the population, such as age.

Answer: As the aim of the study was to map the criteria for the target group as recommended in the German guideline “multimedication”. This guideline does not use an age cut point, therefore we did not apply age as a criterion. The rational behind is that also younger persons might have a high treatment burden and would be therefore benefit from a medication review.

(3) The authors should show the information about database appropriateness, such as generalizability of this database. This journal is an international journal, and the author should provide a short explanation about German insurance system to the readers.

Answer: Thank you for this suggestion. We included some information in the first section, like the fact, that German patients can chose freely their GPs and other specialist in outpatient care, which requires strategies for communication and information flow. (cf. answer to point 14 of the first reviewer).

Besides, we included some information on the statutory health insurance in the method section where we describe the database. We added the following:

About 88% (73 millions) of the total population (83 million) are insured with one of the 103 statutory health insurance funds. Since 1993, there is freedom to choose the insurance fund. The German healthcare system is based on the Bismarck model of a social health insurance system with its principal of solidarity (all insured persons pay a certain percentage of their income regardless of their health status) and self-governance (for an overview cf.[17]) The BARMER is one of the largest health insurers and the data is used since many years for health service research.

In the limitation section we added the following sentences.

Insurance funds differ in their clientele. The BARMER has a somewhat higher percentage of elderly and female insurees, but according to surveys comparable health status [48]. Therefore, we cannot rule out that we overestimate the number of eligible persons due to the age and gender differences

(4) The authors defined for the criteria of multi-morbidity (≥3 chronic diseases) and polypharmacy with 5 or more concurrent active substances (ATC 7-digit) over at least 91 days. Please explain the reason for the cutoff points. 

Answer: As written, the definition of multimorbidity and polypharmacy was driven by the literature. There is no methodological gold standard. Studies of polypharmacy differ not only in terms of the number of drugs but also related to the time-window and the operationalization of polypharmacy (concurrent or cumulative use) To assess the duration of polypharmacy - in order to avoid the inclusion of temporarily poly-medicated -, we analyzed different time windows (3 month, 6 month and more than 9 month) and choose the length of one quarter of a year for the study population. The rationale behind is that multimorbid patients visit the doctors at least once per quarter. The reimbursement of ambulatory care physicians and the documentation of the diagnoses is carried out on a quarterly basis. Besides, the time window of 3 month has also be chosen in other studies (for an overview cf. Guillot J, Maumus-Robert S, Bezin J. Polypharmacy: A general review of definitions, descriptions and determinants. Therapie. 2020;75(5):407-16.). As explanation we added the following sentence:

This time window was chosen to include patients with chronic use of polypharmacy over a substantial period and to exclude patients with short time use.

(5)This study  aims are to describe the frequency of occurrence of some medication safety issues. The authors should analyze and discuss about the risk of adverse effects on the polypharmacy. In addition, in Table 3, the authors should described about something related to the multi-morbidity status and the polypharmacy in this study population. 

Answer: There may be a misunderstanding here related to the aims of our study. We did not aim at analyzing adverse events as an effect of polypharmacy. This would require an analytical study to be carried out. Our aim was to describe how many of those with polypharmacy show a medication pattern with known risk for patient safety. The results of table 3 are those for the defined study population with multimorbidity and polypharmacy.

(6) The recommendation is to  analyze safety issues for the polypharmacy in the multiple model analysis, since only prevalence analysis results in influenced by confounding effects, and it would be not appropriate to conclude this study purpose.

Answer: please see our answer to statement 5

(7) The study limitation is not enough discussed. The authors should discuss about the non-adherence issues.

Answer: We took up this point, that we have no information on non-adherence. According to our understanding this does not affect the analysis, as the patient is still a patient with polypharmacy and (non-)adherence problems should be discussed during the medication review. We added the following sentence:

In addition, we have no information on whether at all (non-adherence) or in what way (irregular) the prescribed drugs have been taken, but we know that the doctors have prescribed these drugs and the patient has collected the medication in a pharmacy and is therefore a patient with intended multimedication.

Further whether claim data has no misclassification error should be discussed.

Answer: We took this point up and added the following sentence to the limitation paragraph.

Misclassification of diagnoses cannot be ruled out in routine data, but for our study the number of diagnostic groups are relevant, not the single diagnosis. Therefore, we are confident that the results are robust.

(8) The gender difference should be observed in terms of the combination of polypharmacy and multi-morbidity issues. 

Answer: We added a sentence concerning gender differences in the study population (fig.1 ) for the age group >64 years.

As especially elderly patients are affected, the age group >64 years is also presented (37.3%); men are more affected (40.5%) compared to women (35.3%).

Round 2

Reviewer 1 Report

The authors revised their manuscript and quite all of my comments were adequately addressed. There are just 2 minor points:

On page 5, zalepon should be corrected to zaleplon.

On page 8, “The above cited study from Spain identified PPI as most prescribed drug with 63,3% […]” should be “63.3%”.

Author Response

Dear Reviewer,

Thank you very much for your time and the hints related to spelling mistakes.

Reviewer 2 Report

The authors revised the comments and this version is appropriate to be published.

Author Response

Dear Reviewer,

thank you very much for your time. We checked the spelling mistakes.